# The Recommended and Excessive Preventive Behaviors during the COVID-19 Pandemic: A Community-Based Online Survey in China

**DOI:** 10.3390/ijerph17196953

**Published:** 2020-09-23

**Authors:** Yisheng Ye, Ruoxi Wang, Da Feng, Ruijun Wu, Zhifei Li, Chengxu Long, Zhanchun Feng, Shangfeng Tang

**Affiliations:** 1School of Medicine and Health Management, Tongji Medical College, Huazhong University of Science and Technology, Wuhan 430030, Hubei, China; yishengye@hust.edu.cn (Y.Y.); ruoxiwang@hust.edu.cn (R.W.); longchengxu@hust.edu.cn (C.L.); 2Research Center for Rural Health Service, Key Research Institute of Humanities & Social Sciences of Hubei Provincial Department of Education, Wuhan 430030, Hubei, China; 3School of Pharmacy, Tongji Medical College, Huazhong University of Science and Technology, Wuhan 430030, Hubei, China; fengda@hust.edu.cn; 4China National Center for Biotechnology Development, Beijing 100039, China; wurj@cncbd.org.cn (R.W.); lizf@cncbd.org.cn (Z.L.)

**Keywords:** COVID-19, health belief model, preventive behaviors, mental health

## Abstract

COVID-19 presents unprecedented challenges to the global public health response. Preventive behaviors and keeping social distance are regarded as compelling ways to prevent COVID-19. This study focused on the sociological and psychological factors associated with proper and excessive preventive behaviors of the COVID-19 outbreak in China. For the sample, we collected the data of 4788 participants who were surveyed between 4 April and 15 April 2020 from eight provinces in China. This study designed a self-filled questionnaire that included demographic information, six components of the Health Belief Model, and target preventive behaviors. Descriptive analysis, Chi-square test, logistic regression analysis, Mantel–Haenszel hierarchical analysis, and propensity score matching were employed in this study. The results showed that 54.7% of the participants had adequate basic prevention, 63.6% of the participants had adequate advanced prevention, and 5.8% of the participants practiced excessive prevention. The elder participants were less likely to engage in proper preventive behaviors. Perceived susceptibility, perceived benefits, perceived barriers, cues to action, and knowledge levels were associated with preventive behaviors. Excessive preventive behaviors in high-risk groups with suspected symptoms were associated with their extreme psychological condition, while the support from the community and family plays an important role in avoiding these behaviors.

## 1. Introduction

Coronavirus disease 2019 (COVID-19) is an infectious disease caused by severe acute respiratory syndrome coronavirus 2. The condition is highly contagious, and its primary clinical symptoms include fever, dry cough, fatigue, muscle aches, and breathing difficulties [1]. Early studies on COVID-19 were mostly related to the pathological, virological, and clinical characteristics of the disease [2,3,4,5,6]. Compared with previous pandemics (such as SARS, MERS, etc.), COVID-19 presents unprecedented challenges to the global public health response, in part, due to its unique epidemiological characteristics, the incubation period of COVID-19 can be up to 24 days [7]. The transmissibility might be higher for COVID-19 than for SARS, and the community spread is more prominent [8], which had led to more overall deaths due to the numerous cases [9]. As of 21 June 2020, COVID-19 had affected more than 8.7 million individuals, resulting in approximately 460,000 deaths [10]. Recent studies have reported that a significant proportion of people infected with COVID-19 did not show any clinical symptoms, which poses a considerable challenge to combat the pandemic [11,12,13]. However, no vaccines or effective antiviral drugs could be used to treat or prevent COVID-19 at this moment [14]. The only dependable measures are the non-drug interventions for public health and psychosocial health issues, which achieved remarkable success in Wuhan [15].

The preventive behaviors and keeping social distance were regarded as compelling ways to prevent disease and promote health, especially in terms of interrupting the transmission of infectious diseases [16]. In China, wearing masks outside, staying at home as much as possible, keeping social distance, and washing hands frequently were recommended as the vital self-protection measures during the pandemic. Although appropriate preventive behaviors can effectively slow the transmission of the virus, the excessive coverage of preventive practices by new social media may cause unnecessary anxiety. Simultaneously, unreasonable or excessive restrictions on activities may incur sedentary behaviors, physical inactivity, increasing health risks, and a range of psychological problems. Previous research has shown that panic and anxiety caused by the pandemic may cause people suffering from some psychological or clinical manifestations, including but not limited to the increasing use of alcohol, Chinese herbal medicines, and other drugs [17]. Along with the virus spreading worldwide, strengthening the public’s preventive actions rationally from the perspective of demographic characteristics, disease cognition, and especially psychology is quite essential for global health.

To the best of our knowledge, most of the previous studies had shown that demographic characteristics are often associated with preventive behaviors during epidemics of infectious diseases such as influenza [18] or COVID-19 [19,20]. Taking this into account, various psychosocial methods, such as the Health Belief Model (HBM), the Stages of Change Model, and the Social Cognition Model, are proposed to predict the practice of preventive behavior at the individual level. The HBM is one of the most widely used models and provides the necessary conceptual framework for this study. It includes perceived severity, perceived susceptibility, perceived benefits, perceived barriers, cues to action, and knowledge levels [21]. Besides, some studies have explored the relationship between disease risk perception and preventive behavior [22], and the psychological impact of COVID-19 on the public [23,24]. However, few existing studies discovered the role of psychological factors in preventive behaviors of COVID-19 [25]. Therefore, this study aims to investigate the preventive behaviors and the differences of practice among different groups. In order to promote residents’ preventive behaviors and provide the precision interventions for countries those were remained stuck in the pandemic or those controlled the prevalence of COVID-19 but faced renewed risk, we also clarified the vulnerable groups of basic prevention and the special psychological state of high-risk groups in this study.

## 2. Materials and Methods

### 2.1. Data Sampling

In this study, participants from the eastern, central, and western regions of China were selected using a directional convenient sampling method, with each region selecting the two provinces with the highest number of patients and one province with the lowest number of patients based on the prevalence of COVID-19 in early April. Therefore, Guangdong, Zhejiang, Fujian, Hunan, Hubei, Shanxi, Sichuan, and Gansu provinces were selected. The details are shown in Figure 1. Since the economic and cultural conditions of Sichuan and Chongqing were similar, only Sichuan Province was selected from the high prevalence group in the western region. In each province, we selected the provincial capitals and another city with the most disease-affected areas in each province, and each city selected 60 families from rural and urban areas in half. To ensure that the residents receiving the survey had the ability to answer questions, the population under the age of 10 were excluded in this study. In the end, a total of 7118 residents from 1920 households in eight provinces participated in this investigation.

Data collection was conducted between 4 April and 15 April 2020. We employed a project manager in each province to coordinate and guide the investigation. Six investigators with university education from the local cities were employed to assist the online investigation after trained by the project manager. Each investigator needs to give a total of 20 electronic questionnaires to their relatives, friends, or familiar local classmates by WeChat and assist in controlling the quality in the survey. Each questionnaire was required to be filled out in 15 min, and then, a small gift was sent out as a reward. Meanwhile, it was recommended to select young and middle-aged participants with strong response willingness for attending the investigation, so that it could ensure a reasonable proportion of the elderly in the sample. If there were individual investigators who could not complete the survey of 20 families, other investigators were assigned to help conduct the supplementary surveys to ensure that the overall number of households met the requirements.

Simultaneously, the process of data collection was effectively controlled through the main measures as follows. (1) Each investigator was an independent individual and trained through a networking meeting. (2) The proportion of people over the age of 60 years was required to reach more than 15% to ensure that the elderly in the family were included in the study as much as possible. (3) Before sending the questionnaire, a unique code for every questionnaire was generated based on the 20 families and their number of members, which meets the survey criteria. (4) After separately sending the questionnaire to the selected families, the participants were told that they would get a small gift as a reward if they answered the questions carefully. Many trap questions were designed to distinguish if the questionnaire qualified or not. (5) The project manager at the provincial level checked the quality of each questionnaire based on the consistency of the two groups and the response time threshold, which is seven and a half minutes.

### 2.2. Measurement

This study designed a self-filled questionnaire with 168 questions based on literature research and previous experience, which mainly includes seven parts: demographic information, physical condition, nutrition and prevention behavior, perception of medical preparation and response, COVID-19 knowledge level, individual health and risk protection, and psychological pressure.

Table 1 shows the details of dependent and independent variables with assignments. The dependent variable in the study was the practice of preventive behaviors. In this study, frequent hand washing and wearing masks were defined as basic preventive behaviors. Compensatory nutrition and exercise were described as advanced preventive behaviors. Basic and advanced preventive behaviors were defined as proper actions. However, the use of traditional Chinese medicine and western medicine for drug therapy was identified as excessive preventive behaviors. This study defined this type of preventive behavior as an overly sensitive preventive behavior because there is no drug to act as a preventive one, which is more likely to be a psychological comfort [17].

In terms of basic preventive behaviors, the measurement was made through two questions, (1) whether respondents wore masks when they went outside, (2) whether respondents had been maintaining personal hygiene practices (such as washing their hands frequently) since the outbreak. The answers of “yes” and “no” were respectively coded as “1” and “0”, the basic prevention was measured by adding up the answers’ score, and a total score of 0 or 1 indicated that basic preventive behavior was deficient, while 2 was adequate.

Advanced preventive behaviors were measured by two questions, (1) whether respondents took the exercise to prevent COVID-19, (2) whether respondents tried their best to intake nutrition to prevent COVID-19. The answers of “yes” and “no” were respectively coded as “1” and “0”. In terms of excessive preventive behavior, it was measured by two questions, (1) whether respondents used Chinese medication to prevent COVID-19, (2) whether respondents used western medicine to prevent COVID-19. The answers of “yes” and “no” were respectively coded as “1” and “0”. The advanced prevention and excessive prevention were measured by adding up the answers’ score, and a total score of 0 or 1 indicates that the preventive behavior is deficient, while 2 is regarded as adequate.

The six components of the health belief model include perceived sensitivity, perceived severity, perceived benefit, perceived barriers, cues to action, and knowledge level. Some separate design questions measure each component.

Three questions measured perceived sensitivity: (1) whether respondents felt vulnerable to COVID-19, (2) whether they had close contact with any relatives or friends infected by COVID-19, and (3) whether they had suspected symptoms, such as a sore throat, dry cough, fever, muscle aches, and fatigue, etc. The answers of “yes” and “no” were respectively coded as “1” and “0”. The perceived sensitivity was measured by adding up the answers’ score. A high score indicated that respondents believed they were highly susceptible to COVID-19 (0 as “not at all,” 1 as “low,” 2 as “middle,” and 3 as “high”).

Perceived severity was measured by three questions, (1) whether respondents were afraid of infection with COVID-19, (2) whether respondents were fearful of dying from COVID-19 infection, and (3) whether respondents had insomnia as they were worried about getting infected. The measurement criteria are the same as the sensitivity of perception. A high score indicates that respondents believe that there are serious adverse consequences after COVID-19 infection (0 as “not at all,” 1 as “low,” 2 as “middle,” and 3 as “high”).

Perceived benefit was judged by the respondents’ attitude towards the effect of preventive measures. They were asked to indicate the degree on a 4-point scale, such as agreeing preventive behaviors could prevent contracting and spreading COVID-19 (0 represents strongly disagree, 3 represents strongly agree).

Perceived barriers were measured by difficulty in obtaining a mask and disinfectant, difficulty in getting medication from a pharmacy, and difficulty in shopping at the supermarket. Respondents answered “yes” or “no” and then added up the yes responses to form an overall score. A high score indicates high barriers to preventive behavior (0 was regarded as “not at all”, and 3 was regarded as “high”).

Cues to action were measured by three items: support from family and friends, government, and experts. Respondents answered “yes” or “no” and then added up the yes responses to form an overall score. A high score indicates that respondents are encouraged to engage in preventive behavior (0, 1, 2, 3 were regarded as “not at all”, “a few”, ”average”, and “a lot”, respectively).

Knowledge level was measured using four basic questions related to COVID-19. (1) Washing hands and wearing a mask frequently could help to prevent the COVID-19, (2) when the infected patients sneeze or cough around people, is it very easier to infect them, (3) eating lots of garlic could not help prevent COVID-19, and (4) improving immunity could help fight COVID-19. The answers “yes” and “no” were respectively coded as “1” and “0”. The knowledge level was measured by adding up the answers’ score. The higher scores indicated higher level of knowledge such as 1 means “answering one question correctly”, and 4 means “answering four questions correctly”.

### 2.3. Statistical Analysis

Descriptive analysis, Chi-square test, binary logistic regression analysis, Mantel–Haenszel hierarchical analysis, and propensity score matching were employed in this study. All variables were represented by frequency distribution and percentage, and descriptive analysis was conducted on demographic characteristics and other variables. Chi-square tests were performed to compare the correlation between different social demographic characteristics and dependent variables. Those variables significantly related to preventive behaviors in the univariate analysis could finally be included in a binary logistic regression model. A backward stepwise regression analysis was adopted to discover and eliminate the factors that have little influence on the model.

The demographic variables and six components of the HBM were set as independent variables, and the deficient basic and advanced preventive behaviors were respectively set as the outcome variables. To further identify the vulnerable groups with a high risk of being affected by the six components of the HBM, Mantel–Haenszel hierarchical analysis of deficient basic prevention was conducted across sub-populations. This stratified analysis is a commonly used method for controlling confounders [26]. It stratifies data according to the confounders that need to be controlled and then estimates the association between exposure/treatment factors and study outcomes.

In order to assess the differences in the psychological health states, it was necessary to distinguish the variations between the intervention group (group B) with suspected symptoms and the intervention group (group A) without suspected symptoms. There were confounding factors, such as age, sex, and income in both groups, misleading to the conclusions. Therefore, propensity score matching (PSM) was used to balance the demographic characteristics of the two groups [27]. In the control group, individuals with the same initial personal characteristics as those with suspected symptoms were matched. The correlation between the two groups was balanced and became statistically indistinguishable after matching. The associations between independent variables and prevention behaviors were analyzed by calculating odds ratios (ORs) and 95% confidence interval (CI). All data were performed by using the SPSS 16.0 software (IBM Corp. Armonk, NY, USA). A *p* value less than 0.05 was considered statistically significant.

### 2.4. Ethics

The protocol was reviewed. Ethical approval was obtained from the Ethics Committee of Tongji Medical College, Huazhong University of Science and Technology (2020S107). The oral informed consent was obtained from each participant before taking the online survey.

## 3. Results

### 3.1. Characteristics of Participants

A total of 4788 participants were included in this analysis. Approximately a third of the participants were aged 21 to 40 and 41 to 60. More than half of the participants were women, and 59.5% of the participants were married. About one-fifth of the participants were students, and a minority (5.7%) of the participants were incapacity of work. Nearly half of the participants’ annual household income is less than CNY 100,000; more than two-fifths of the participants had university education, and 45.8% of the participants live in the central area; the overwhelming majority of the participants lived in cities with others.

### 3.2. The Prevalence of the Basic, Advanced, and Excessive Preventive Behaviors

As the distributions of land for basic, advanced, and excessive preventive practices are shown in Figure 2, approximately 54.7% of the participants had adequate basic preventive behaviors, while 11.4% of the participants failed to implement any essential preventive action. As for the implementation of advanced preventive practices, more than three-fifths (63.6%) of the participants performed well. Concerning excessive preventive behaviors, only 5.8% of the participants had adequate excessive preventive behaviors.

### 3.3. Differences in the Adoption of Basic, Advanced, and Excessive Preventive Behavior

Table 2 summarizes the adoption of basic, advanced, and excessive preventive behavior overall and differences among different subgroups. Age, gender, occupation, and education were associated with the adoption of basic and advanced preventive behaviors. The number of suspected symptoms were associated with all behaviors’ adoption. The percentage of basic preventive behavior’s adoption was seemingly higher among those who were aged between 21 and 40, worked in a big company, and had a master’s level education. For advanced preventive behavior’s adoption, the proportion was seemly higher among those who were aged below 20, worked in government or public institutions, and had a college education. In terms of excessive preventive behavior’s adoption, the percentage was higher among those who had more than one suspected symptom.

The participants who were aged over 60, divorced or widowed but not married, worked as farmer/fisherman/herdsman, and had less than 6 years of education were less likely to engage in basic and advanced preventive behaviors.

In terms of family variables, living areas and living places were associated with the adoption of basic and advanced preventive behaviors. Adopting advanced preventive behaviors were also associated with household income and lifestyles. Interestingly, people living in eastern China had the lowest proportion of basic preventive behaviors and the highest proportion of advanced preventive behaviors. In contrast, those living in western China showed the opposite, but all the percentages of these behaviors were over 50%. The percentage of basic and advanced preventive behaviors was higher among urban residents than among rural residents. For, the proportion of advanced preventive behavior was seemly higher among those residents with household income between CNY 300,000–400,000 and lived with others. The details were shown in Table 3.

### 3.4. Factors Associated with the Adoption of Basic, Advanced, and Excessive Preventive Behaviors

In terms of basic preventive behaviors, the respondents who lived in western China (OR = 1.405, 95%CI = 1.199–1.645) and had a higher education (OR = 2.453, 95%CI = 1.830–3.289) were more likely to adopt basic preventive practice. Besides, the respondents who were aged over 60 years old (OR = 0.770, 95%CI = 0.623–0.951); lived in rural areas (OR = 0.567, 95%CI = 0.503–0.639); were divorced (OR = 0.515, 95%CI = 0.383–0.692); and worked as farmer, fisherman, or herdsman (OR = 0.377, 95%CI = 0.273–0.522) were less likely to adopt basic prevention.

As for advanced preventive behaviors, the respondents who worked in government or public institution (OR = 1.390, 95%CI = 1.042–1.856), were students (OR = 1.356, 95%CI = 1.042–1.766), and had a higher education (OR = 2.266, 95%CI = 1.687–3.043) were more likely to perform the advanced prevention. In addition, the respondents who were aged over 60 years old (OR = 0.528, 95%CI = 0.423–0.659), lived in rural areas (OR = 0.714, 95%CI = 0.631–0.808), lived alone (OR = 0.699, 95%CI = 0.566–0.864), and did not have a job (OR = 0.664, 95%CI = 0.476–0.926) were less likely to adopt advanced prevention.

The respondents who were elderly (OR = 1.716, 95%CI = 1.028–2.867) were more likely to perform excessive preventive behaviors. In summary, the respondents who were aged over 60, lived in rural areas, were divorced, and did not have a job were more likely to ignore the essential preventive practices. More details were shown in Table 4.

The results of the six components of the Health Belief Model are presented in Table 5. In terms of the recommended prevention, the respondents who felt more susceptible to be infected with COVID-19 (OR = 1.636, 95%CI = 1.060–2.525), believed that the infection had more severe consequences (OR = 1.296, 95%CI = 1.066–1.576), believed in the greater benefits of preventive behaviors (OR = 6.007, 95%CI = 2.401–15.029) and who had higher knowledge level (OR = 3.149, 95%CI = 1.165–8.510) were more likely to adopt the essential preventive practice. The respondents who were elderly (OR = 1.716, 95%CI = 1.028–2.867), felt more susceptible to be infected with COVID-19 (OR = 4.390, 95%CI = 2.293–8.430), and had a higher knowledge level (OR = 1.659, 95%CI = 1.266–2.174) were more likely to adopt the excessive preventive behaviors. Perceived barriers (OR = 0.829, 0.713–0.966) were the risk factors of basic and advanced preventive practices, while cues to action (OR = 0.313, 95%CI = 0.099–0.992) were risk factors of excessive preventive behaviors.

The study also identified the vulnerable populations including the respondents who were aged above 60 years old (OR = 1.66, 95%CI = 1.08–2.57), had a low-income level (OR = 1.75, 95%CI = 1.19–2.56), with a household income < CNY 100,000 (OR = 1.91, 95%CI = 1.35–2.72), no work or no work ability (OR = 3.49, 95%CI = 1.49–8.18), and had junior high school education (OR = 2.34, 95%CI = 1.48–3.69). They are more likely to perceive barriers while performing preventive actions, resulting in deficient basic prevention. The details were shown in Figure 3.

### 3.5. Differences of the Psychological Health States between Symptomatic and Asymptomatic Populations

Table 6 presents the results of the propensity score matching. Before matching, there were 874 residents in the intervention group and 3844 residents in the control group. After matching, 874 residents were in both the intervention group and the control group. The differences in occupation (*p* = 0.01), household income (*p* < 0.001), and living place (*p* = 0.003) between the control group and the intervention group were statistically significant before matching. There was no statistically significant difference in demographic factors such as age, gender, and income between the groups after matching (*p* > 0.05), so the matching was relatively acceptable. However, the difference of psychological health states between the groups was statistically significant (*p* < 0.01).

Table 7 showed a further analysis of the paired population, at which time confounders such as gender and income have been well balanced between the control group and the intervention group. The overall psychological condition of the patients with suspected symptoms was significantly worse than that of healthy people (*p* < 0.01). They were more likely to have a range of psychological problems, manifested in more severe depression, helplessness, and loneliness (*p* < 0.01).

## 4. Discussion

In a comparison of young populations, the elderly practiced less virus prevention. The implementation rate (46.6%) is considerably lower than the rate observed in a previous study, which was also conducted in China [28]. Some aspects of their attitude may explain the lower chance of practice. First, although wearing mask was suggested as a mandatory precaution [28,29], their effectiveness in preventing COVID-19 is still very controversial in the population [30,31]. Similarly, a prior study suggested that older people might be reluctant to accept advice from the Centers for Disease Control to wear masks [32]. Second, due to the reduced physical function and adverse consequences [33], older people have a lower intention to engage in social activities. Lack of social interaction might affect the practice of preventive behaviors [22]. Third, the elderly have lower education levels than other populations. Previous studies had found that people with lower education levels reported poorer hygiene habits [34,35] and weaker awareness of self-care [36]. They are also more likely to acquire health knowledge from social media, where information accuracy and quality are doubtful [37,38]. Older age has been confirmed as a significant independent predictor of mortality in COVID-19 [5]. The case fatality rate in older age groups is substantially higher than that in younger groups, with an increasing profile with age [39]. Moreover, the clinical symptoms of the elderly after infection are more severe [5]. Therefore, the low level of practice of preventive behavior among older population deserves the government and public health institutions’ great concern.

This study also revealed that the perceived sensitivity, perceived severity, perceived benefits, cues to action, and knowledge levels are substantial predictors accounting for the practice of prevention behaviors, which are in line with the previous literature [40,41]. It showed higher ORs for preventive behavior with perceived sensitivity; that is, people are more likely to take feasible preventive actions to avoid infection if there are confirmed cases in their community [7]. Another critical thing to note is that perceived barriers was a significant factor impeding the implementation of preventive behaviors, which is consistent with existing evidence [40]. It might relate to the shortage of masks and goods in supermarkets and pharmacies in China in February [28]. What is more, some preventive behavior, such as wearing masks, might be relatively difficult to perform due to discomfort, inconvenience, and adverse skin reactions [42].

Furthermore, those who were aged over 60 years old, had a low-income level, with an annual household income less than CNY 100,000, with no work or no workability, and had junior high school education belong to the vulnerable groups. They were more likely to perceive the barriers to preventive behaviors than other groups, which in turn leads to inadequate basic preventive behaviors. Therefore, the most important thing for the government is to strengthen the stock and supply of materials, ensuring that residents have access to necessities such as food and masks. In addition to providing materials to the general public, it is necessary to provide targeted assistance to vulnerable groups, such as delivering free masks for the elderly and low-income people door-to-door.

Previous retrospective studies have also reported that perceived benefits tend to have a greater motivational effect [43]. Importantly, those with higher perceived benefits had an almost a 3-times increased likelihood of exercise or absorbing nutrition and had more than a 6-times increased likelihood of wearing masks or washing hands compared with those with low perceived benefits. There are several possible explanations. First, as we all know, there were no drugs to treat the disease [14]. Second, wearing masks and handwashing remains one of the most effective ways to prevent contracting and spreading COVID-19 [41,44,45]. Third, numerous epidemiological studies have shown that improving the individual’s immunity can reduce the risk of infection and the harm caused by the disease [46,47]. Thus, the belief in benefits will lead individuals to take preventive measures. However, a prior study indicated that the awareness of benefits is uneven as the epidemic develops [48]. This condition may be related to the weak ability to understand information and the lack of available channels for receiving information among these subgroups such as rural residents [37,38].

Concerning health education, the government should give priority to using new media such as the internet to improve public health literacy and promote the practice of preventive behaviors, especially conduct the more effective health education interventions for target groups [49]. Given that vulnerable people have difficulty perceiving information by new media, a diagrammatic or video format in simple languages were suggested to ensure the coverage. Additionally, we found that a high level of perceived sensitivity and knowledge level were associated with over-sensitive preventive behaviors. Those who believed they were highly susceptible had an over 4-fold increased likelihood of taking drugs to prevent COVID-19 than other groups, which may be associated with more suspected symptoms in this population. Developing the analysis, we found that after controlling demographic variables, the psychological condition of those with suspected symptoms was significantly worse than that in healthy population. They are more likely to develop a range of psychological problems [50], which are manifested as health concerns, frequent feelings of depression, helplessness, and loneliness, resulting in an extreme mental state. Similarly, a prior study showed that abnormal psychological conditions in the COVID-19 pandemic might lead to inappropriate use of drugs [17]. The higher level of disease knowledge might enhance the individual’s concern with their health [24,28], which in turn deepens this stress state.

We hypothesized that although they knew taking drugs would not directly prevent disease, they could relieve their psychological burden with symptomatic relief. Hence, this might yield the overly sensitive preventive behavior of taking drugs at the first sign of suspected symptoms. Importantly, we identified cues for action as a practical approach to curb this behavior. Prior evidence showed that support from family members and health experts can contribute significantly to practicing the appropriate preventive behaviors [21,40,41]. Family and community support, care, and professional advice might be good for relieving the emotional, psychological stress, and help to avoid stressful actions. In terms of psychological counseling, local health facilities must first identify high-risk population with suspected symptoms. Then, facilities should provide psychological guidance through mobile phones or websites to avoid virus transmission [51]. Secondly, families should pay attention to their mental state and timely communicate with them. We could place some posters in prominent locations to guide the residents to follow appropriate behaviors.

## 5. Limitations

This study has several limitations. Firstly, the study used electronic questionnaires for data collection. Those who had not access to the internet were not adequately investigated. Better-educated people may make up too much of the study sample, and they might be more willing to commit to preventive behavior. Secondly, the cross-sectional data limits our ability to verify and demonstrate causal relationships between variables. Studies that collect data at multiple points in the pandemic may yield different results. Thirdly, although the social expectations deviation of anonymous online surveys was lower than the telephone or face-to-face survey, there was still some possibility of reaction bias in the electronic questionnaire. Forth, the preventive behaviors and psychological factors involved in this study may not be comprehensive. For example, the numbers of daily handwashing and self-efficacy instances were not included in the study. Future studies may explore the impact of interventions in the preceding paragraph on preventive behaviors and the psychological conditions. Fifth, respondents may have a large subjective bias in answering specific questions such as wearing masks and washing their hands, so the accuracy of the study will be affected to some extent. Future research will consider detailing each of these questions.

## 6. Conclusions

This study estimated the differences between and influencing factors on different groups’ implementation of prevention behaviors. The implementation rate of proper preventive behaviors is relatively low. The younger participants with higher education were associated with a higher implementation rate of proper behaviors. Perceived sensitivity, perceived severity, perceived benefits, cues to action, and knowledge levels are the predictors of proper preventive behavior. Perceived sensitivity and knowledge levels are the predictors of excessive preventive behavior. We also found that over-prevention behaviors in high-risk groups with suspected symptoms were associated with their extreme psychological condition. The support from community and family plays a significant role in controlling dreadful behaviors. Although this is a cross-sectional study, and we cannot determine causation, these findings provide strong evidence to establish a relationship between the psychological condition and excessive preventive behaviors.

## Figures and Tables

**Figure 1 ijerph-17-06953-f001:**
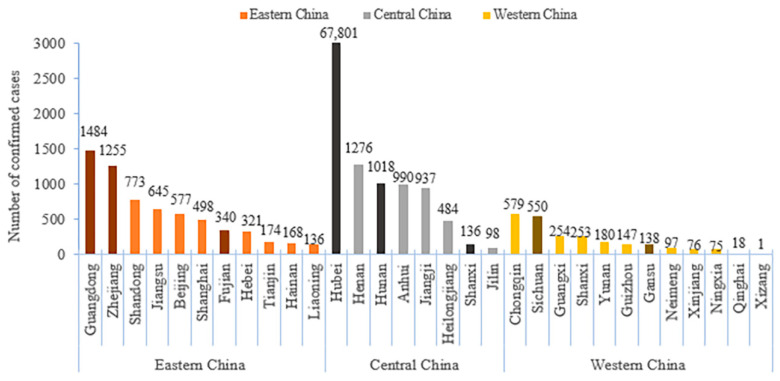
The provincial prevalence of COVID-2019 on 1 April 2020 in Eastern, Central, and Western China.

**Figure 2 ijerph-17-06953-f002:**
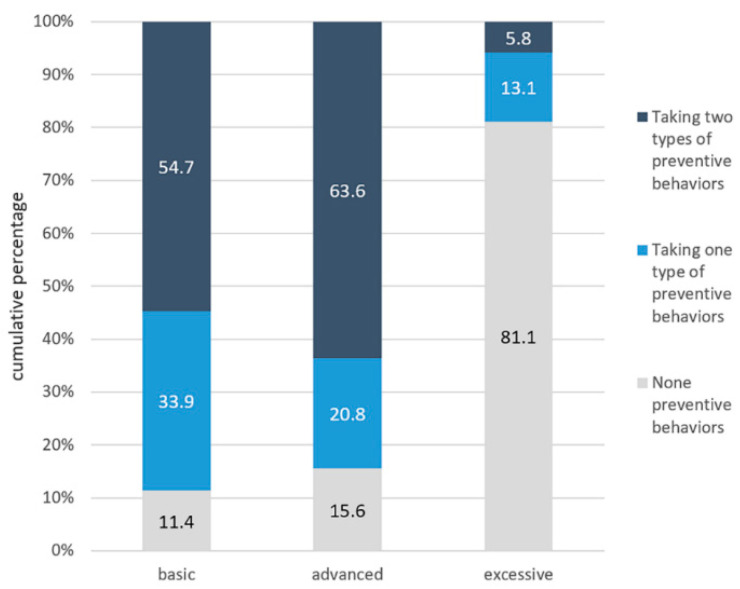
The cumulative percentage of the performance of prevention behaviors.

**Figure 3 ijerph-17-06953-f003:**
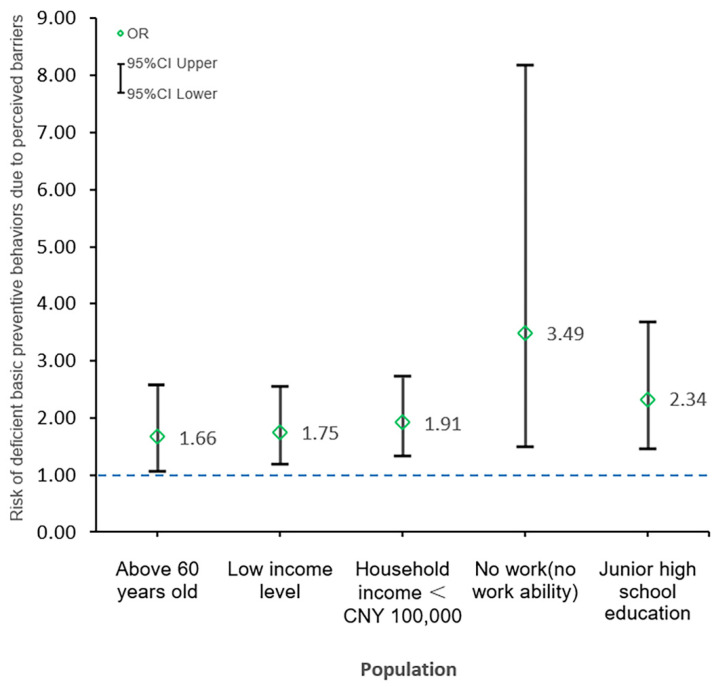
The vulnerable populations and a higher risk of deficient basic prevention.

**Table 1 ijerph-17-06953-t001:** Variables and assignments.

Variables	Assignments
Dependent variables:	
Basic prevention	0 = no; 1 = yes
Advanced prevention	0 = no; 1 = yes
Excessive prevention	0 = no; 1 = yes
The Health Belief Model:	
Perceived sensitivity	0 = not at all; 1 = low; 2 = medium; 3 = high
Perceived severity	0 = not at all; 1 = low; 2 = medium; 3 = high
Perceived benefit	0 = strongly disagree; 1 = disagree; 2 = agree; 3 = strongly agree
Perceived barriers	0 = not at all; 1 = low; 2 = medium; 3 = high
Cues to action	0 = not at all; 1 = a few; 2 = average; 3 = a lot
Knowledge levels	1 = answering one question correctly; 2 = answering two questions correctly; 3 = answering three questions correctly; 4 = answering four questions correctly

**Table 2 ijerph-17-06953-t002:** Differences in the adoption of basic, advanced, and excessive preventive behaviors among participants with different individual variables.

Variables	Total	Basic Prevention	Advanced Prevention	Excessive Prevention
	**No. (%)**	**No. (%)**	**No. (%)**	**No. (%)**
**Total**	4788 (100)	2621	3043	277
**Age (years)**				
<20	599 (12.5)	318 (53.1)	414 (69.1)	22 (3.7)
21–40	1774 (37.1)	1057 (59.6)	1164 (65.6)	113 (6.4)
41–60	1601 (33.4)	867 (54.2)	1024 (64.0)	92 (5.7)
>60	814 (17.0)	379 (46.6)	441 (54.2)	50 (6.1)
*p-Value*		<0.001	<0.001	0.102
**Gender**				
Male	2248 (47.0)	1203 (53.5)	1412 (62.8)	144 (6.4)
Female	2540 (53.0)	1418 (55.8)	1631 (64.2)	133 (5.2)
*p-Value*		0.109	0.315	0.084
**Marriage status**				
Unmarried	1725 (36.0)	973 (56.4)	1160 (67.2)	87 (5.0)
Married	2851 (59.5)	1565 (54.9)	1783 (62.5)	178 (6.2)
Divorced	212 (4.5)	83 (39.2)	100 (47.2)	12 (5.7)
*p-Value*		<0.001	<0.001	0.241
**Occupation**				
Waiting for employment	300 (6.3)	180 (60)	185 (61.7)	22 (7.3)
No work (no work ability)	273 (5.7)	123 (45.1)	141 (51.6)	17 (6.2)
Self-employed shop owner or entrepreneurs	569 (11.9)	305 (53.6)	361 (63.4)	33 (5.8)
Staff in government or public institution	615 (12.8)	370 (60.2)	425 (69.1)	33 (5.4)
Famer/fisherman/herdsman	321 (6.7)	116 (36.1)	164 (51.1)	21 (6.5)
Retired	499 (10.4)	278 (55.7)	303 (60.7)	33 (6.6)
students	1155 (24.1)	604 (52.3)	792 (68.6)	55 (4.8)
Staff in big company	276 (5.8)	189 (68.5)	187 (67.8)	18 (6.5)
Staff in a middle or small company	426 (8.9)	256 (60.1)	281 (66.0)	22 (5.2)
The others	354 (7.4)	200 (56.5)	204 (57.4)	23 (6.5)
*p-Value*		<0.001	<0.001	0.769
**Education**				
<6 years	698 (14.6)	284 (40.7)	337 (48.3)	36 (5.2)
7–9years	809 (16.9)	413 (51.1)	484 (59.8)	39 (4.8)
10–12years	865 (18.1)	505 (58.4)	574 (66.4)	53 (6.1)
13–16years	2145 (44.8)	1253 (58.4)	1464 (68.3)	129 (6.0)
>16years	271 (5.6)	166 (61.3)	184 (67.9)	20 (7.4)
*p-Value*		<0.001	<0.001	0.472
**Number of suspected symptoms**				
0	3914 (81.7)	2207 (56.4)	2544 (65.0)	227 (5.8)
1	382 (8.0)	172 (45.0)	221 (57.9)	10 (2.6)
2	252 (5.3)	131 (52.0)	149 (59.1)	21 (8.3)
>2	240 (5.0)	111 (46.3)	129 (53.8)	19 (7.9)
*p-Value*		<0.001	<0.001	0.007

**Table 3 ijerph-17-06953-t003:** Differences in the adoption of basic, advanced, and excessive preventive behaviors among participants with different family variables.

Variables	Total	Basic Prevention	Advanced Prevention	Excessive Prevention
	**No. (%)**	**No. (%)**	**No. (%)**	**No. (%)**
**Total**	4788 (100)	2621	3043	277
**Household income**				
<CNY 100,000	2074 (43.3)	1148 (55.4)	1248 (60.2)	121 (5.8)
CNY 100,000–200,000	1735 (36.2)	929 (53.5)	1130 (65.1)	101 (5.8)
CNY 200,000–300,000	579 (12.2)	323 (55.8)	396 (68.4)	33 (5.7)
CNY 300,000–400,000	193 (4.0)	105 (54.4)	136 (70.5)	13 (6.7)
>CNY 400,000	207 (4.3)	116 (56.0)	133 (64.3)	9 (4.3)
*p-Value*		0.787	<0.001	0.890
**Living Areas**				
Eastern China	1317 (27.5)	685 (52.0)	878 (66.7)	66 (5.0)
Central China	2191 (45.8)	1182 (53.9)	1373 (62.7)	135 (6.2)
Western China	1280 (26.7)	754 (58.9)	792 (61.9)	76 (5.9)
*p-Value*		0.001	0.020	0.355
**Living Place**				
Urban	3065 (64.0)	1829 (59.7)	2033 (66.3)	173 (5.6)
Rural	1723 (36.0)	792 (46.0)	1010 (58.6)	104 (6.0)
*p-Value*		<0.001	<0.001	0.577
**Living style**				
Living with others	4370 (91.3)	2386 (54.6)	2802 (64.1)	248 (5.7)
Living alone	418 (8.7)	235 (56.2)	241 (57.7)	29 (6.9)
*p-Value*		0.525	0.009	0.291

**Table 4 ijerph-17-06953-t004:** Binary logistic regression analysis between demographic factors and the adoption of basic, advanced, and excessive preventive behavior.

Variables	Basic Preventive Behaviors	Advanced Preventive Behaviors	Excessive Preventive Behaviors
	OR (95% CI)	OR (95% CI)	OR (95% CI)
**Age (refer to below 20)**			
21–40	1.303 * (1.081–1.570)	0.853 (0.699–1.041)	1.784 * (1.119–2.845)
41–60	1.044 (0.865–1.260)	0.793 * (0.649–0.970)	1.599 (0.994–2.571)
>60	0.770 * (0.623–0.951)	0.528 ** (0.423–0.659)	1.716 * (1.028–2.867)
**Areas (refer to eastern China)**			
Central China	1.125 (0.979–1.293)	0.878 (0.759–1.015)	1.215 (0.897–1.646)
Western China	1.405 ** (1.199–1.645)	0.836 * (0.711–0.984)	1.194 (0.849–1.679)
**Living in rural areas (refer to urban)**	0.567 ** (0.503–0.639)	0.714 ** (0.631–0.808)	1.087 (0.844–1.399)
**Living alone (refer to living together)**	0.976 (0.790–1.205)	0.699 * (0.566–0.864)	1.350 (0.895–2.036)
**Marriage status (refer to unmarried)**			
Married	0.941 (0.833–1.064)	0.770 ** (0.676–0.877)	1.297 (0.989–1.702)
Divorced	0.515 ** (0.383–0.692)	0.449 ** (0.336–0.600)	1.098 (0.589–2.048)
**Occupation (refer to waiting for employment)**			
No work (no work ability)	0.547 ** (0.392–0.762)	0.664 * (0.476–0.926)	0.839 (0.436–1.616)
Self-employed	0.770 (0.580–1.023)	1.079 (0.808–1.440)	0.778 (0.445–1.360)
Staff in government OR public institution	1.007 (0.760–1.335)	1.390 * (1.042–1.856)	0.716 (0.410–1.252)
Famer/fisherman/herdsman	0.377 ** (0.273–0.522)	0.649 * (0.472–0.894)	0.885 (0.476–1.644)
Retired	0.839 (0.627–1.122)	0.961 (0.716–1.289)	0.895 (0.511–1.566)
students	0.731 * (0.564–0.946)	1.356 * (1.042–1.766)	0.632 (0.379–1.054)
Staff in big company	1.448 * (1.027–2.041)	1.306 (0.927–1.841)	0.882 (0.462–1.681)
Staff in a middle or small company	1.004 (0.743–1.357)	1.205 (0.886–1.638)	0.688 (0.374–1.267)
Others	0.866 (0.634–1.183)	0.845 (0.618–1.157)	0.878 (0.479–1.609)
**Exact household income in 2019 (refer to the level of >CNY 400,000**			
<CNY 100,000	1.106 (0.826–1.481)	0.946 (0.699–1.279)	1.451 (0.723–2.914)
CNY100,000–200,000	0.926 (0.690–1.241)	1.057 (0.780–1.432)	1.399 (0.695–2.817)
CNY200,000–300,000	0.973 (0.705–1.344)	1.177 (0.841–1.648)	1.336 (0.627–2.845)
CNY300,000–400,000	0.875 (0.588–1.302)	1.249 (0.818–1.906)	1.537 (0.641–3.685)
**Education (refer to less than 6 years)**			
7–9 years	1.552 ** (1.264–1.905)	1.595 ** (1.300–1.957)	0.931 (0.585–1.482)
10–12 years	2.128 ** (1.733–2.611)	2.113 ** (1.722–2.593)	1.200 (0.776–1.855)
13–16 years	2.147 ** (1.798–2.564)	2.303 ** (1.935–2.741)	1.177 (0.805–1.720)
over 16 years	2.453 ** (1.830–3.289)	2.266 ** (1.687–3.043)	1.465 (0.832–2.580)

Note: OR = odds ratio; CI = confidence interval; * *p* < 0.05, ** *p* < 0.01.

**Table 5 ijerph-17-06953-t005:** Binary logistic regression analysis between six factors of the Health Belief Model and the adoption of basic, advanced, and excessive preventive behavior.

Variables	Basic Preventive Behaviors	Advanced Preventive Behaviors	Excessive Preventive Behaviors
	OR (95% CI)	OR (95% CI)	OR (95% CI)
**Perceived sensitivity** **(refer to 0)**			
1	1.252 * (1.074–1.460)	0.991 (0.845–1.163)	1.493 * (1.025–2.178)
2	1.160 (0.970–1.387)	0.923 (0.767–1.110)	2.307 ** (1.551–3.432)
3	1.636 * (1.060–2.525)	1.325 (0.841–2.089)	4.390 ** (2.293–8.430)
**Perceived severity** **(refer to not at all)**			
Low	0.952 (0.655–1.358)	1.155 (0.921–1.448)	1.212 (0.760–1.934)
Middle	0.993 (0.798–1.236)	1.027 (0.847–1.143)	1.196 (0.713–1.681)
High	1.296 * (1.066–1.576)	1.085 (0.888–1.326)	0.961 (0.626–1.473)
**Perceived benefits** **(refer to strongly disagree)**			
Disagree	2.259 (0.819–6.229)	0.555 (0.227–1.360)	0.131 (0.012–1.424)
Agree	2.912 * (1.144–7.412)	1.635 (0.754–3.542)	0.173 (0.011–2.638)
strongly agree	6.007 ** (2.401–15.029)	2.883 * (1.357–6.127)	0.218 (0.015–3.257)
**Perceived barriers** **(refer to not at all)**			
Low	0.636 ** (0.540–0.749)	0.910 (0.769–1.077)	0.776 (0.538–1.120)
Middle	0.649 ** (0.542–0.777)	0.966 (0.801–1.164)	0.878 (0.597–1.292)
High	0.829 * (0.713–0.966)	0.854 * (0.731–0.997)	1.096 (0.807–1.487)
**Cues to action** **(refer to not at all)**			
A few	0.485 (0.296–1.239)	1.332 (0.431–4.114)	0.074 * (0.008–0.713)
Average	0.632 (0.457–1.352)	1.264 (0.460–3.475)	0.130 * (0.034–0.497)
A lot	0.769 (0.304–1.950)	2.984 * (1.111–8.014)	0.313 * (0.099–0.992)
**Knowledge levels** **(refer to answering one question correctly)**			
Answering two questions correctly	1.165 * (1.004–1.352)	1.997 (0.469–5.369)	1.254 (0.413–3.807)
Answering three questions correctly	1.442 (0.692–3.005)	0.821 (0.389–1.733)	1.391 (0.823–2.681)
Answering four questions correctly	3.149 * (1.165–8.510)	1.011 (0.869–1.177)	1.659 ** (1.266–2.174)

Note: OR = odds ratio; CI = confidence interval; * *p* < 0.05, ** *p* < 0.01.

**Table 6 ijerph-17-06953-t006:** Differences in demographic characteristics and psychological health states between two groups before and after using propensity score matching.

Variables	Group	Before Matching			After Matching		
		(n = 4718)			(n = 1748)		
		Mean	T Value	*p*-Value	Mean	T Value	*p*-Value
Age	A	40.71	−1.519	0.129	41.71	−0.048	0.962
	B	41.76			41.76		
Gender	A	1.53	−0.701	0.483	1.55	0.336	0.737
	B	1.54			1.54		
Marry status	A	1.68	−0.012	0.990	1.69	0.336	0.737
	B	1.68			1.68		
Occupation	A	5.74	2.578	0.010	5.71	1.79	0.074
	B	5.5			5.5		
Individual income	A	2.2	−0.828	0.408	2.18	−1.084	0.279
	B	2.23			2.23		
Household income	A	1.86	−5.203	<0.001	1.99	−1.436	0.151
	B	2.06			2.06		
Education	A	3.09	−1.478	0.139	3.08	−1.252	0.211
	B	3.15			3.15		
Living place	A	0.37	3.004	0.003	0.34	1.12	0.263
	B	0.32			0.32		
Psychological health states	A	2.28	−10.115	<0.001	2.25	−8.593	<0.001
	B	2.5			2.5		

**Table 7 ijerph-17-06953-t007:** Differences in the psychological health states between symptomatic and asymptomatic populations with different variables.

Variables	Option	Asymptomatic Group	Symptomatic Group	χ2	*p*-Value
		No (%)	No (%)		
Psychology health states	Better	30 (54.5)	25 (54.5)	79.91	<0.001
	Same	609 (57.9)	442 (42.1)		
	Little worse	220 (38.4)	353 (61.6)		
	Far worse	15 (21.7)	54 (78.3)		
Feeling depressed	Same	520 (58.5)	369 (41.5)	65.6	<0.001
	Little worse	330 (43.5)	429 (56.5)		
	Far worse	24 (24.0)	76 (76.0)		
Feeling helpless	Same	617 (56.0)	485 (44.0)	46.78	<0.001
	Little worse	236 (41.2)	337 (58.8)		
	Far worse	21 (28.8)	52 (71.2)		
Feeling lonely	Same	632 (54.8)	522 (45.2)	38.50	<0.001
	Little worse	216 (43.2)	284 (56.8)		
	Far worse	26 (27.7)	68 (72.3)		

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
