# Peer review of "The Recommended and Excessive Preventive Behaviors during the COVID-19 Pandemic: A Community-Based Online Survey in China"

_ijerph, 2020, doi:10.3390/ijerph17196953_

Round 1

Reviewer 1 Report

This is a very interesting study, which is a survey of the actual situation and attitudes regarding the prevention of the people in China under the COVID-19 pandemic. However, as is usual in such questionnaire surveys, if there are no clear criteria for participants to answer, the subjective bias of participants may increase. Therefore, I have one comment.

Comment. For example, in “Basic preventive behaviors”, two questions, (1) whether respondents wore masks when they go outside, 2) whether respondents had been maintaining personal hygiene practices (such as wash hands frequently) since the outbreak, were performed, and the participants answered "yes" or "no". However, on wearing a mask when going out, if the participant think that "I generally wore it but did not wear it several times", which should the participant answer "yes" or "no"? “Whether to answer with" depends on the subjectivity of the participants, therefore, I think it lacks accuracy. The question "maintaining personal hygiene practices (such as wash hands frequently)" is also inaccurate because the way of receiving “frequently” will differ depending on the participants. I think it is better to clarify that there is a possibility that the participants' subjective bias may be large, including other question items.

Author Response

Response to Reviewer 1 Comments

Point 1:

This is a very interesting study, which is a survey of the actual situation and attitudes regarding the prevention of the people in China under the COVID-19 pandemic. However, as is usual in such questionnaire surveys, if there are no clear criteria for participants to answer, the subjective bias of participants may increase. Therefore, I have one comment.

Comment. For example, in “Basic preventive behaviors”, two questions, (1) whether respondents wore masks when they go outside, 2) whether respondents had been maintaining personal hygiene practices (such as wash hands frequently) since the outbreak, were performed, and the participants answered "yes" or "no". However, on wearing a mask when going out, if the participant think that "I generally wore it but did not wear it several times", which should the participant answer "yes" or "no"? “Whether to answer with" depends on the subjectivity of the participants, therefore, I think it lacks accuracy.

Response 1:

Thanks for your comments. This is our previous oversight. Apologize sincerely. We have added a fifth point to the limitations and propose that there is a certain subjectivity in the respondents' answers. (Page 14 Line 417-419 in the clean revision)

In response to your suggestions, our future research will make modifications to this point. Wearing masks will be subdivided into the following five categories: never, seldom, sometimes, often, and always [1].

Never: not wearing a face mask within 10 times

Seldom: wearing a face mask 1–3 times within 10 times

Sometimes: wearing a face mask 4–6 times within 10 times

Often: wearing a face mask 7–9 times within 10 times

Always: wearing a face mask 10 times within 10 times

  1. Lee, L.Y.-k.; Lam, E.P.-w.; Chan, C.-k.; Chan, S.-y.; Chiu, M.-k.; Chong, W.-h.; Chu, K.-w.; Hon, M.-s.; Kwan, L.-k.; Tsang, K.-l., et al. Practice and technique of using face mask amongst adults in the community: a cross-sectional descriptive study. Bmc Public Health 2020, 20, doi:10.1186/s12889-020-09087-5.

Point 2:

The question "maintaining personal hygiene practices (such as wash hands frequently)" is also inaccurate because the way of receiving “frequently” will differ depending on the participants. I think it is better to clarify that there is a possibility that the participants' subjective bias may be large, including other question items.

Response 2:

Thanks for your comments. We have added a fifth point to the limitations that there was subjective bias among participants. (Page 14 Line 417-419)

Future studies will consider directly consulting participants about the number of times they wash their hands each day.

Reviewer 2 Report

Dear authors, 

I consider your manuscript as a very actual one. Anyway, please find my comments below:

  • abstract: It should be written as one text (without "Results:");
  • Introduction: At the end of this section add the structure of your manuscript (it would be presented for readers);
  • Figure 1: Add the name of the Y-axis;
  • Structure: The manuscript is divided into many short sections (see 2.2.2, 2.2.3, 2.3.1). I recommend to change this content and create more extensive sections;
  • Methods: Descriptive analysis, Chi-square test, binary logistic regression analysis, and Mantel-haenszel hierarchical analysis, backward stepwise regression, ... were used. These methods are poorly described. It could be better-explained why did you choose these methods. Add also previous their usages (or combinations). There are frequently-used methods, so you are able to with it lightly;
  • Figure 2: Legend (0, 1,2) is not described;
  • Table 2, Table 3: I recommend to divide these tables into 2-3 tables (each of them) and better describe these results (which are very interesting for the readers);
  • Figure 3: The quality of this figure is very low. The name of the X-axis is also missing;
  • Table 4, Table 5: Remove the text "Table 5 showed a further..." between these tables and better describe them.

I wish all the best with this one and other manuscripts.  

Author Response

Response to Reviewer 2 Comments

Point 1:

I consider your manuscript as a very actual one. Anyway, please find my comments below:

abstract: It should be written as one text (without "Results:");

Response 1: Thank you for your comments sincerely. We have rewritten the abstract as one text, as you commanded. (Page 1 Line 18-33 in the clean revision)

Point 2:

Introduction: At the end of this section add the structure of your manuscript (it would be presented for readers);  

Response 2: Thank you for your comments sincerely. We have added the structure of our manuscript at the end of the introduction. (Page 2 Line 76-81)

Point 3:

Figure 1: Add the name of the Y-axis;  

Response 3: Thank you for your comments sincerely. We have added the name of the Y-axis for figure 1. The name was “Numbers of confirmed cases”. (Page 3 Line 95-96)

Point 4:

Structure: The manuscript is divided into many short sections (see 2.2.2, 2.2.3, 2.3.1). I recommend to change this content and create more extensive sections; 

Response 4: Thank you for your comments sincerely. We have changed this content and created more extensive sections. There are no longer too many short sections in the article now. (Page 4 Line 134 - Page 5 Line 184)

Point 5:

Methods: Descriptive analysis, Chi-square test, binary logistic regression analysis, and Mantel-haenszel hierarchical analysis, backward stepwise regression, ... were used. These methods are poorly described. It could be better-explained why did you choose these methods. Add also previous their usages (or combinations). There are frequently-used methods, so you are able to with it lightly;

Response 5: Thank you for your comments sincerely.

  1. Descriptive analysis, Chi-square test, and binary logistic regression analysis are frequently-used methods, so we did not have much explanation.

  1. In our study, backward stepwise regression is a variable selection method in the process of binary logistic regression analysis. This method begins with a model that contains all variables under consideration, then starts removing the least significant variables one after the other. The process iterates until a pre-specified stopping rule is reached or until no variable is left in the model. We have described this method in more detail. (Page 5 Line 193-194)

  1. Then for further analysis, methods such as Mantel-Haenszel Hierarchical Analysis were used. We have explained the reason why we choose these methods, and cited the corresponding literature [1,2] to describe their usual usages.

3.1 Mantel-haenszel hierarchical analysis (Page 5 Line 197-201)

[1].  Tang, S.; Bishwajit, G.; Ji, L.; Feng, D.; Fang, H.; Fu, H.; Shao, T.; Shao, P.; Liu, C.; Feng, Z., et al. Improving the Blood Pressure Control With the ProActive Attitude of Hypertensive Patients Seeking Follow-up Services Evidence From China. Medicine 2016, 95, doi:10.1097/md.0000000000003233.

3.2 Propensity score matching (Page 5 Line 204-209)

[2].   Cunningham, S.A.; Adams, S.R.; Schmittdiel, J.A.; Ali, M.K. Incidence of diabetes after a partner's diagnosis. Preventive Medicine 2017, 105, 52-57, doi:10.1016/j.ypmed.2017.08.020.

Point 6:

Figure 2: Legend (0, 1,2) is not described;

Response 6: Thank you for your comments sincerely. We have described the legend for figure2. The original legend (0,1,2) is now described as (none preventive behaviors, taking one type of preventive behaviors, taking two types of preventive behaviors).

 (Page 6 Line 232-234)

Point 7:

Table 2, Table 3: I recommend to divide these tables into 2-3 tables (each of them) and better describe these results (which are very interesting for the readers);

Response 7: Thank you for your comments.

We have divided Table 2 into 2 tables (Table 2 and Table 3)and described the results in more detail (Page 6 Line 237- Page 7 Line 247 & Page 8 Line 251-259)

We also have divided Table 3 into 2 tables (Table 4 and Table 5) as you recommend and described the results in more detail (Page 8 Line 263-277 & Page 9 284 - Page 10 Line 294)

Point 8:

Figure 3: The quality of this figure is very low. The name of the X-axis is also missing;

Response 8: Thank you for your comments. We have redesigned this figure and added the X-axis. The name of the X-axis is “population”. (Page 11 Line 307)

Point 9:

Table 4, Table 5: Remove the text "Table 5 showed a further..." between these tables and better describe them. I wish all the best with this one and other manuscripts.

Response 9: Thank you for your comments sincerely. We have removed the text "Table 5 showed a further..."  and describe more details for original Table 4 and Table 5. (Page 11 Line 310-317 & Page 12 Line 320-324)